# The Prognostic Value of Postoperative Radiotherapy for Thymoma and Thymic Carcinoma: A Propensity-Matched Study Based on SEER Database

**DOI:** 10.3390/cancers14194938

**Published:** 2022-10-08

**Authors:** Chi Zhang, Qin Wang, Liwen Hu, Zhuangzhuang Cong, Yong Qiang, Fei Xu, Zheng Zhang, Chao Luo, Bingmei Qiu, Xiaokun Li, Yi Shen

**Affiliations:** 1Department of Cardiothoracic Surgery, Jinling Hospital, Medical School of Nanjing University, Nanjing 210008, China; 2Department of Cardiothoracic Surgery, Jinling Hospital, School of Medicine, Southeast University, Nanjing 210096, China; 3Department of Cardiothoracic Surgery, Jinling Hospital, School of Clinical Medicine, Nanjing Medical University, Nanjing 210029, China; 4Department of Cardiothoracic Surgery, Jinling Hospital, School of Clinical Medicine, Southern Medical University, Guangzhou 510515, China; 5Department of Cardiothoracic Surgery, Jinling Hospital, Nanjing 210000, China; 6Department of Thoracic Surgery, West China Hospital, Sichuan University, Chengdu 610044, China

**Keywords:** postoperative radiotherapy, PORT, thymoma, thymic carcinoma, SEER program, survival

## Abstract

**Simple Summary:**

Although surgery has been recognized as the cornerstone of treatment for patients with resectable thymic epithelial tumors, the role of postoperative radiotherapy remains controversial. We performed this SEER-based propensity-matched analysis to investigate the prognostic value of postoperative radiotherapy in thymoma and thymic carcinoma. The results showed that postoperative radiotherapy improved both overall survival and cancer-specific survival in patients with Masaoka-Koga stage IIB–IV thymoma. This study is the first to demonstrate the prognostic value of postoperative radiotherapy in stage IIB thymic carcinoma. This large, up-to-date population-based longitudinal study may provide guidance on the use of postoperative radiotherapy for a thymoma or thymic carcinoma.

**Abstract:**

(1) Objectives: The effect of postoperative radiotherapy (PORT) for thymoma and thymic carcinoma remains controversial. This study aimed to investigate the prognostic value of PORT for thymoma and thymic carcinoma in a population-based registry. (2) Methods: This retrospective study used the Surveillance, Epidemiology, and End Results (SEER) database to identify patients diagnosed with thymoma and thymic carcinoma between 2010 and 2019. Propensity score matching was performed to adjust statistical influences between the PORT and non-PORT groups. (3) Results: A total of 2558 patients with thymoma (*n* = 2138) or thymic carcinoma (*n* = 420) were included. In the multivariate analysis, PORT was an independent prognostic factor for OS (overall survival; p < 0.001) and CSS (cancer-specific survival; *p* = 0.001) in thymoma and an independent prognostic factor for OS in thymic carcinoma (*p* = 0.018). Subgroup analyses revealed that PORT was beneficial to OS and CSS in patients with Masaoka-Koga stage IIB-IV thymoma (OS: IIB, *p* < 0.001; III-IV, *p* = 0.005; CSS: IIB, *p* = 0.015; III-IV, *p* = 0.002) and stage IIB thymic carcinoma (OS: *p* = 0.012; CSS: *p* = 0.029). (4) Conclusion: This propensity-matched analysis identified the prognostic value of PORT in thymoma and thymic carcinoma based on the SEER database. For patients with stage IIB-IV thymoma and stage IIB thymic carcinoma, PORT was associated with improved OS and CSS. A more positive attitude towards the use of PORT for nonlocalized thymoma and thymic carcinoma may be appropriate.

## 1. Introduction

Thymoma and thymic carcinoma are the most common anterior mediastinal thymic epithelial tumors, although they are still relatively rare in general [1]. Before the World Health Organization (WHO) Consensus Committee published the distinction between the diagnosis and histological features of thymoma and thymic carcinoma in 1999, they were often confused [2]. Thymoma is a potentially malignant disease that can invade mediastinal organs and is associated with many autoimmune paraneoplastic diseases [3,4,5]. Thymic carcinoma is more advanced than thymoma and has a worse prognosis, with lymphatic or hematogenous metastasis in about 30% of cases [6,7].

The Masaoka staging system was first proposed by Masaoka et al. in 1981 [8]. In 1994, Koga et al. modified this system and proposed the Masaoka-Koga staging system [9], which is now widely accepted as the clinical staging standard for thymic epithelial tumors. Thymic epithelial tumors are classified into stages I to IV based on the local extension of the primary tumor and the degree of involvement of the surrounding organs [10]. When feasible, surgical resection is the recommended treatment for thymoma and thymic carcinoma, and the extent of resection proved to be an independent predictor of survival [11]. In addition, postoperative radiotherapy (PORT) is also considered an important component of treating thymic epithelial tumors. Still, there is no consensus on its optimal use, especially for Masaoka-Koga stage II patients [12,13,14]. According to the guidelines of the National Comprehensive Cancer Network (NCCN) and the European Society for Medical Oncology (EMSO), PORT is recommended for incompletely resected thymoma and thymic carcinoma and for Masaoka-Koga stage III-IV completely resected thymoma and thymic carcinoma [15,16]. The low incidence of thymic epithelial tumors made randomized trials difficult to conduct and hindered the development of evidence-based recommendations.

The National Cancer Institute’s (NCI) Surveillance, Epidemiology, and End Results (SEER) database is a nationwide cancer dataset that tracks demographic and clinical data for nearly one-third of the United States population. This study investigated the prognostic value of PORT on patients with thymoma and thymic carcinoma using the SEER database. Considering the selection bias for the receipt of PORT in the database, propensity score matching was performed to balance the distribution of baseline clinicopathological variables.

## 2. Methods

### 2.1. Database

This retrospective study analyzed the SEER 18-Registry maintained by the NCI (1975–2019; dataset submitted in November 2021; www.seer.cancer.gov (accessed on 16 August 2022)). The SEER*Stat software (version 8.3.9; National Institutes of Health, Bethesda, MD, USA) was used to extract clinicopathologic and survival information from the database. Permission to access the research data file in the SEER registry was received from the NCI, USA (reference No. 16521-Nov 2021).

### 2.2. Study Population

Patients with a primary site of the thymus (C37.9) between 1975 and 2019 were initially identified. The inclusion criteria were as follows: (1) histologic types of thymoma (8580–8585) and thymic carcinoma (8023, 8033, 8070, 8082, 8123, 8140, 8200, 8260, 8310, 8430, 8480, 8560, 8576, 8586, 8588, and 8589) with the malignant behavior code (/3) according to the International Classification of Disease for Oncology, Third Edition (ICD-O-3) [17]; (2) age >18 years; (3) diagnosis between 2010 and 2019; (4) receipt of cancer-directed surgery with or without PORT. Demographic features and clinicopathological characteristics of these patients were collected, such as the age of diagnosis, gender, race, year of diagnosis, other malignancies, time from diagnosis to treatment, tumor size, lymph node dissection, the extent of surgery, WHO classification, and histological grade. Due to a lack of information on the Masaoka-Koga stages in SEER, we inferred stages based on the primary tumor extension: stage I-IIA (localized; confined to the gland of origin, not otherwise specified), stage IIB (regional; invasion to the adjacent connective tissue), stage III-IV (distant; invasion to the adjacent organs/structures or pleural/pericardial implants and metastases), and unknown (unknown extent of disease). This approach was previously used by Fernandes et al. [18] and Mou et al. [19] to assign the Masaoka-Koga stages, but stages I and IIA or III and IV could not be distinguished based on the SEER data.

### 2.3. Study Outcomes

The outcomes of the present study were overall survival (OS) and cancer-specific survival (CSS). OS was measured from the date of diagnosis to the date of death from any cause. CSS was measured from the date of diagnosis to the date of death directly or indirectly from thymic epithelial tumors. Survival status was shown as “Vital Status” in the SEER database.

### 2.4. Propensity Score Matching

Selection bias due to baseline characteristics in the database may affect the receipt of PORT. A propensity score, the probability of being assigned to the PORT or non-PORT groups given the clinicopathological baseline, was performed to minimize selection bias. The propensity scores were developed from the non-parsimonious logistic regression model with baseline covariates consisting of age of diagnosis, gender, race, year of diagnosis, other malignancies, time from diagnosis to treatment, Masaoka-Koga stage, tumor size, lymph node dissection, the extent of surgery, WHO classification, and histological grade. The PORT groups were matched to non-PORT groups using 1:1 matching based on the nearest neighbor method with a caliper width of 0.02.

### 2.5. Statistical Analysis

Categorical variables were analyzed using Pearson’s chi-square test or Fisher’s exact test, and continuous variables were analyzed using Student’s *t*-test. The Kaplan–Meier method was used to calculate the OS and CSS curves before and after propensity score matching between PORT and non-PORT groups. The log-rank test was performed to determine statistical significance. Univariate Cox regression analysis was used to determine variables associated with the OS and CSS of the matched population, and variables with a *p*-value less than 0.1 were selected for the multivariate Cox regression model. Hazard ratios (HR) with 95% confidence intervals (CI) were reported, and two-sided *p*-values less than 0.05 indicated statistical significance. All statistical analyses were performed using SPSS (version 27.0, SPSS Inc. Chicago, IL, USA).

## 3. Results

### 3.1. Baseline Characteristics

A total of 2558 patients with thymoma (*n* = 2138) or thymic carcinoma (*n* = 420) were identified from the SEER database based on the eligibility criteria. Baseline characteristics are summarized in Table 1. Among the study population, the mean age ± standard deviation (SD) for patients with thymoma was 59.7 ± 14.1 years (range, 19–94), and that for patients with thymic carcinoma was 63.1 ± 12.9 years (range, 19–92). The median tumor size was 6.5 cm for thymoma and 6.0 cm for thymic carcinoma. Postoperative radiotherapy was performed in 963 (45.0%) thymoma and 168 (40.0%) thymic carcinoma patients. Based on the staging method (described in the Study Population), 909 (35.5%), 1181 (46.2%), 394 (15.4%), and 74 (2.9%) patients were classified as Masaoka-Koga stage I-IIA, IIB, III-IV and unknown, respectively. The vast majority of patients were in the localized (I-IIA) or regional (IIB) stage, and thymic carcinoma had a higher proportion of the distant stage (III-IV) than thymoma (20.7 and 14.4%, respectively). In addition, thymic carcinoma had a higher proportion of patients histologically graded poor or undifferentiated than thymoma (25.0 and 8.9%, respectively), and more than half (56.2%) underwent lymph node dissection. More patients with thymoma underwent total or radical resection than patients with thymic carcinoma (58.3 and 51.9%, respectively).

### 3.2. Survival Outcomes before and after Propensity Score Matching

Table 2 and Table 3 show the balances of each variable before and after propensity score matching in thymoma and thymic carcinoma. In the matched cohort, there were 783 thymoma patients each in the PORT group and non-PORT group and 156 thymic carcinoma patients each in the PORT group and non-PORT group. The *p*-values for all covariates after matching were more than 0.1, indicating that the propensity score matching minimized potential selection bias in PORT reception.

In the overall cohort, the average follow-up periods were 94.1 months (95% CI, 92.1–96.4 months, Kaplan–Meier estimate) for thymoma and 80.8 months (95% CI, 75.4–86.1 months) for thymic carcinoma. The Kaplan–Meier survival curves of the overall cohort according to the receipt of PORT are shown in Figure 1. For patients with thymoma, the PORT group had a better OS than the non-PORT group (five-year survival rates were 80.8 and 78.0%, *p* = 0.021, respectively; Figure 1a, but PORT was not associated with CSS before matching (87.3 and 88.4%, *p* = 0.819; Figure 1b. For patients with thymic carcinoma, both the OS (69.0 and 58.2%, *p* = 0.002; Figure 1c) and CSS (77.1 and 67.6%, *p* = 0.030; Figure 1d) of the PORT group were better than the non-PORT group.

In the matched cohort, the Kaplan–Meier survival curves of the OS and CSS for thymoma stratified by PORT or non-PORT are shown in Figure 2a and Figure 3a, respectively. Both the OS (81.3 and 74.7%, *p* < 0.001) and CSS (88.2 and 84.4%, *p* = 0.026) of the PORT group were better than the non-PORT group after matching. The Kaplan–Meier survival curves of the OS and CSS for thymic carcinoma are shown in Figure 4a and Figure 5a, respectively. There were no significant differences in the OS and CSS between the PORT and non-PORT groups in thymic carcinomas (*p* > 0.05).

Subgroup survival analysis of the OS and CSS was performed stratified by the Masaoka-Koga stage (Figure 2b–d, Figure 3b–d, Figure 4b–d and Figure 5b–d). Among the patients with stage IIB and III/IV thymoma, PORT was associated with a better OS (IIB: 81.6 and 75.2%, *p* < 0.001; III/IV: 65.5 and 45.6%, *p* = 0.005, respectively) and CSS (IIB: 89.7 and 85.8%, *p* = 0.015; III/IV: 74.6 and 53.7%, *p* = 0.002, respectively) than non-PORT. However, for patients with stage I/IIA thymoma, PORT had no benefit on the OS (91.6% vs. 90.8%, *p* = 0.415) and even had a negative effect on the CSS (95.4% vs. 97.2%, *p* = 0.042). PORT was beneficial only for the stage IIB thymic carcinoma (OS: 71.5% vs. 61.0%, *p* = 0.012; CSS: 79.9% vs. 69.6%, *p* = 0.029), but not for other stages (*p* > 0.05). For patients with stage III-IV thymic carcinoma, PORT did not significantly improve their survival (OS: 37.9% vs. 23.3%, *p* = 0.149; CSS: 48.0% vs. 34.0%, *p* = 0.293).

### 3.3. Cox Regression Analysis

Table 4 and Table 5 list 11 variables included in the univariate Cox regression model after propensity score matching for thymoma and thymic carcinoma, respectively. Variables with a univariate analysis *p* < 0.1 were selected for the multivariate Cox regression models. The results of the multivariate Cox regression analysis showed that PORT was an independent prognostic factor for both the OS (*p* < 0.001) and CSS (*p* = 0.001) in patients with thymoma. Besides, age, Masaoka–Koga stage, tumor size, and histological grade were also independent prognostic factors for both the OS and CSS. Gender was an independent predictor for OS (*p* = 0.011) but not CSS (*p* = 0.109). For thymic carcinoma, the Masaoka–Koga stage and extent of surgery were independent prognostic factors for both the OS and CSS. PORT was an independent prognostic factor for OS (*p* = 0.018), and tumor size was an independent prognostic factor for CSS (*p* = 0.021).

### 3.4. Subgroup Analysis by Forest Plot

Hazard ratios with 95% CIs for the OS and CSS in the prespecified subgroups are shown in Figure 6 (thymoma, Figure 6a,b; thymic carcinoma, Figure 6c,d). For patients with Masaoka-Koga stage IIB and III/IV thymoma, the PORT was a favorable factor for the OS (HR: 1.79, 95% CI: 1.30–2.47 and HR: 1.73, 95% CI: 1.17–2.55, respectively) or CSS (HR: 1.72, 95% CI: 1.10–2.67 and HR: 2.08, 95% CI: 1.30–3.33, respectively). However, for patients with stage I/IIA thymoma, PORT had no benefit on the OS (HR: 0.78, 95% CI: 0.44–1.41) and was an unfavorable factor for CSS (HR: 0.38, 95% CI: 0.15–1.00). For patients with Masaoka-Koga stage IIB thymic carcinoma, PORT was a favorable factor for the OS (HR: 2.30, 95% CI: 1.18–4.48) or CSS (HR: 2.36, 95% CI: 1.07–5.21). PORT had no benefit on the OS and CSS in patients with stage I-IIA and III-IV thymic carcinoma.

## 4. Discussion

Although surgery has been recognized as the cornerstone of treatment for patients with resectable thymic epithelial tumors, the role of PORT remains controversial [12,20]. In this population-based study, we analyzed the survival outcomes of thymoma and thymic carcinoma patients over the last decade using data from the SEER database. We found that PORT was an independent prognostic factor for both OS and CSS in patients with thymoma and an independent prognostic factor for OS in patients with thymic carcinoma after propensity score matching. Patients with thymoma and thymic carcinoma who received PORT had better OS before and after matching. In subgroup analysis, PORT improved both OS and CSS in patients with Masaoka-Koga stage IIB–IV thymoma and Masaoka-Koga stage IIB thymic carcinoma.

For stage I thymoma, several studies have demonstrated excellent local control rates with surgery. Forquer et al. [21] divided patients into localized (stage I) and regional (stage II-III) groups, and results showed that PORT had no advantage in patients with stage I thymoma and thymic carcinoma (5-year CSS rate: 91% vs. 98%, *p* = 0.03). A randomized trial of 29 patients with stage I thymoma by Zhang et al. [22] showed no difference in outcomes between PORT and surgery alone. Whereas a study based on the International Thymic Malignancy Interest Group (ITMIG) and the European Society of Thoracic Surgeons (ESTS) found that PORT improved the OS and RFS in patients with stage I thymic carcinoma [6], some studies showed no benefit from PORT in this setting [23,24]. Our results showed that both the OS and CSS were shorter in the PORT group for patients with stage I/IIA thymoma or thymic carcinoma. Due to a lack of information in the SEER database, stage I and IIA could not be distinguished, so it was impossible to conclude whether stage IIA patients would benefit from PORT. Nonetheless, we noticed that the results seemed to support the view that complete resection was sufficient for stage I thymoma and thymic carcinoma patients.

The use of PORT is most controversial in stage II thymoma and thymic carcinoma. According to the ESMO guidelines [15], PORT is not recommended after complete resection of stage II thymoma but can be considered in the setting of aggressive histology (type B2, B3) or extensive transcapsular invasion (stage IIB). For stage II thymic carcinoma, the ESMO guidelines indicate that PORT should be considered. A study based on the National Cancer Data Base (NCDB) by Jackson et al. [24] reported that PORT improves OS in patients with stage IIB thymoma (HR = 0.61, *p* = 0.035). Still, no difference was observed in patients with stage I-IIA thymoma or thymic carcinoma. A recent meta-analysis including 4746 patients revealed that PORT was associated with a significantly better OS for patients with stage II (HR = 0.63, 95% CI: 0.44–0.91, *p* = 0.01) and stage III (HR = 0.72, 95% CI: 0.55–0.95, *p* = 0.02) thymoma [25]. This study is the first to demonstrate the prognostic value of PORT in stage IIB thymic carcinoma. We found that patients with stage IIB thymic carcinoma undergoing PORT had better OS (71.5% vs. 61.0%, *p* = 0.012) and CSS (79.9% vs. 69.6%, *p* = 0.029) than surgery alone.

It is now generally accepted that PORT is of great value in improving the prognosis of patients with stage III-IV thymoma and thymic carcinoma, with or without complete resection [6,7,15,26]. Our study also confirmed the positive effect of PORT on the OS and CSS in patients with stage III-IV thymoma. However, for patients with stage III-IV thymic carcinoma, there was no significant difference in survival between the PORT and non-PORT groups (OS: *p* = 0.149; CSS: *p* = 0.293).

In addition to PORT, we also found that age, gender, Masaoka-Koga stage, tumor size, and histological grade were independent prognostic factors for thymoma. Multi-institutional studies have confirmed the significant impact of the Masaoka-Koga stage on thymoma prognosis [7,27,28]. It is worth mentioning that female patients had a better prognosis than male patients (HR = 0.759, *p* = 0.015), contrary to the previous SEER-based study of thymic neuroendocrine tumors (TNETs) [29]. One explanation may be that the proportion of male patients with Masaoka stage IIB or III-IV was more in this study. In addition, many studies reported a slight predominance of women with type A, AB, and B1, which might be one of the reasons [15]. Our study also found that the extent of resection was an independent prognostic factor for patients with thymic carcinoma. Many studies have confirmed that complete resection is an important prognostic factor for resectable thymic carcinoma [30,31].

Previously, several SEER-based studies investigated the prognostic value of PORT in patients with thymoma or thymic carcinoma [19,32,33,34]. However, we differed from previous studies in that (1) propensity matching was performed to minimize selection bias, (2) the latest data from 2010 to 2019 were included, (3) subgroup analysis was performed stratified by the Masaoka-Koga stage, and (4) simultaneous study of thymoma and thymic carcinoma. This large, up-to-date population-based longitudinal study may provide guidance on the use of PORT for thymoma or thymic carcinoma.

This study had several limitations. First, there is a lack of information about Masaoka-Koga stages and radiotherapy regimens in the SEER database. Second, selection bias cannot be completely avoided in this retrospective study. Third, due to the ethnic diversity of the SEER database, our results may not apply to regions with high levels of homogeneity.

## 5. Conclusions

In conclusion, this propensity-matched analysis identified the prognostic value of PORT in thymoma and thymic carcinoma based on the SEER database (2010–2019). For patients with Masaoka-Koga stage IIB-IV thymoma and stage IIB thymic carcinoma, PORT was associated with improved OS and CSS. Further randomized controlled trials are needed to determine the efficacy and indications of PORT for thymoma and thymic carcinoma. A more positive attitude towards the use of PORT for nonlocalized thymoma and thymic carcinoma may be appropriate.

## Figures and Tables

**Figure 1 cancers-14-04938-f001:**
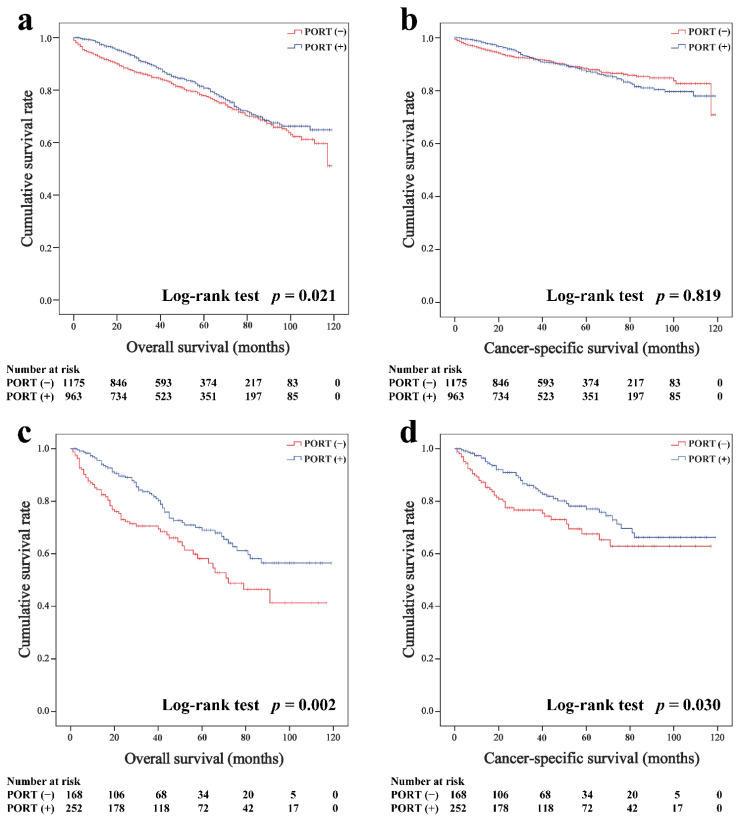
Overall and cancer-specific survival in patients with thymoma (**a**,**b**) and thymic carcinoma (**c**,**d**) before matched.

**Figure 2 cancers-14-04938-f002:**
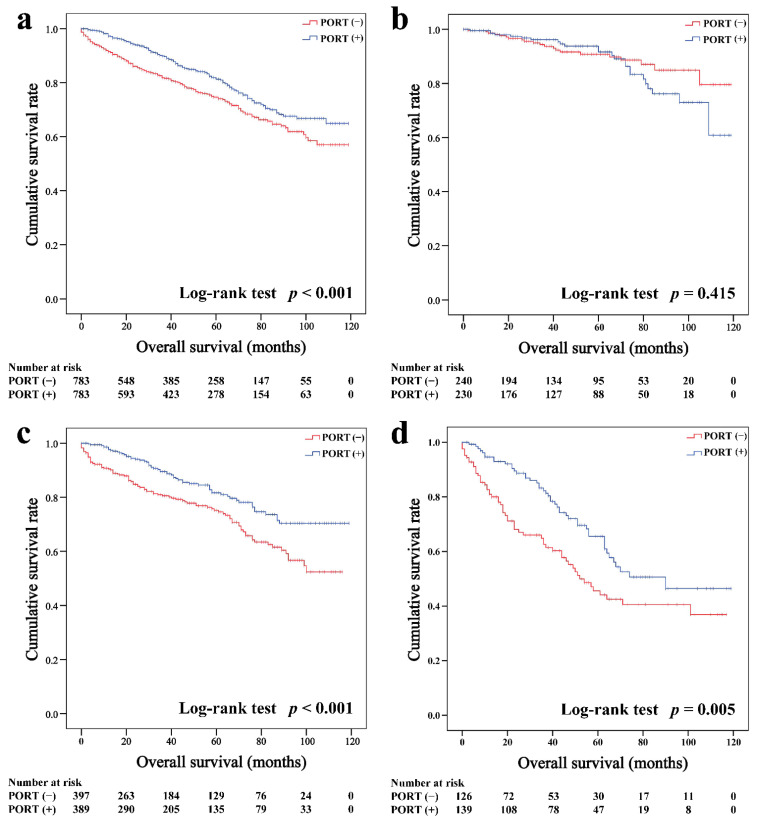
Overall survival in all (**a**), Masaoka–Koga stage I-IIA (**b**), IIB (**c**), and III-IV (**d**) thymoma patients.

**Figure 3 cancers-14-04938-f003:**
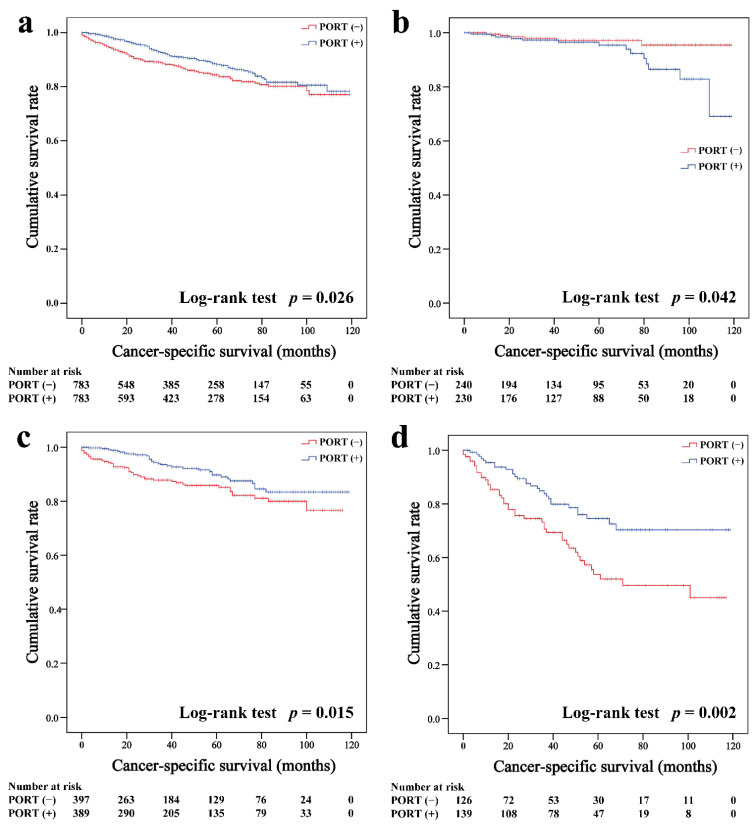
Cancer-specific survival in all (**a**), Masaoka–Koga stage I-IIA (**b**), IIB (**c**), and III-IV (**d**) thymoma patients.

**Figure 4 cancers-14-04938-f004:**
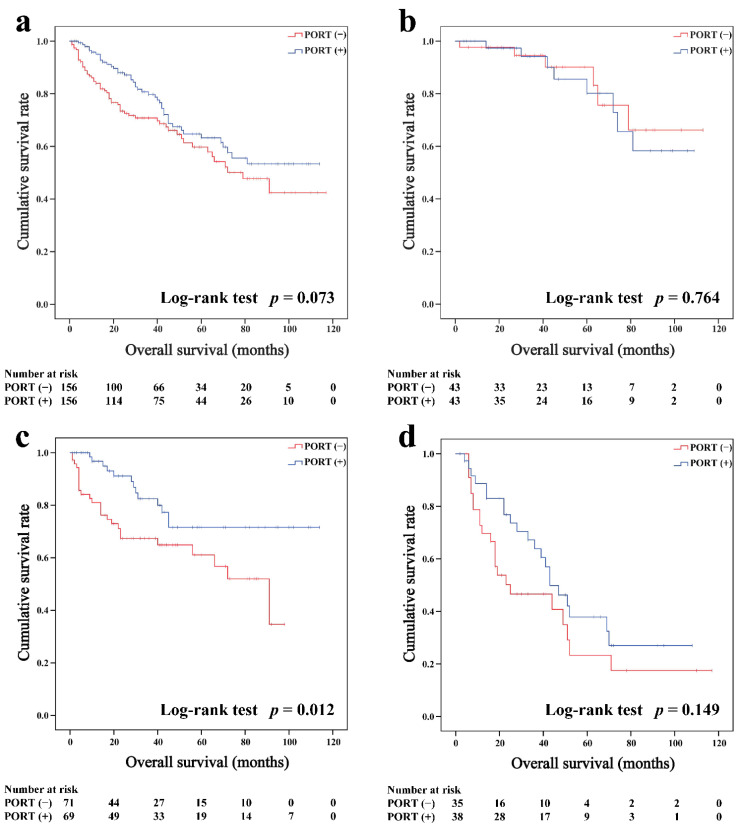
Overall survival in all (**a**), Masaoka–Koga stage I-IIA (**b**), IIB (**c**), and III-IV (**d**) thymic carcinoma patients.

**Figure 5 cancers-14-04938-f005:**
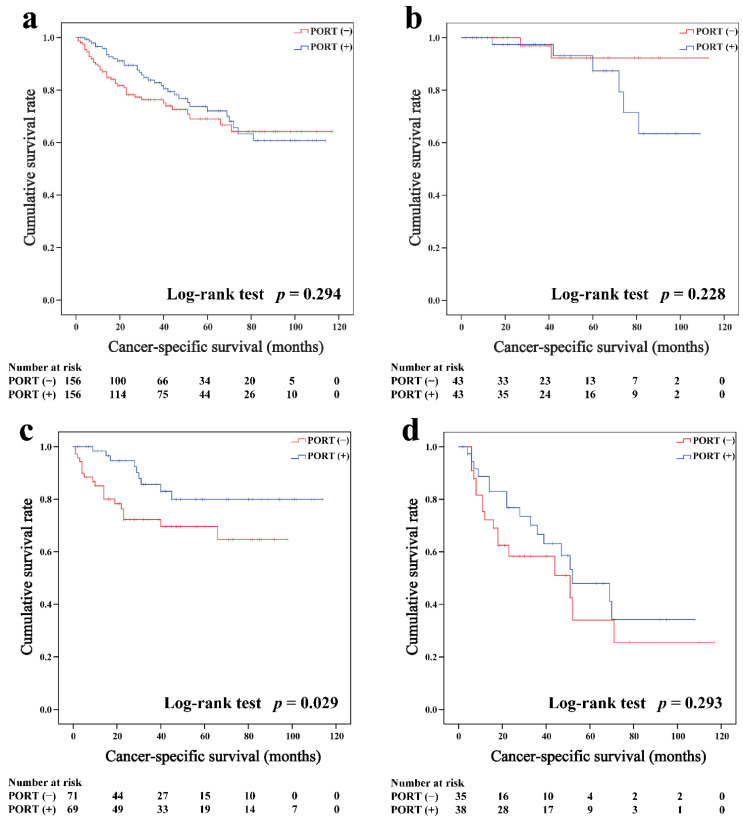
Cancer-specific survival in all (**a**), Masaoka–Koga stage I-IIA (**b**), IIB (**c**), and III-IV (**d**) thymic carcinoma patients.

**Figure 6 cancers-14-04938-f006:**
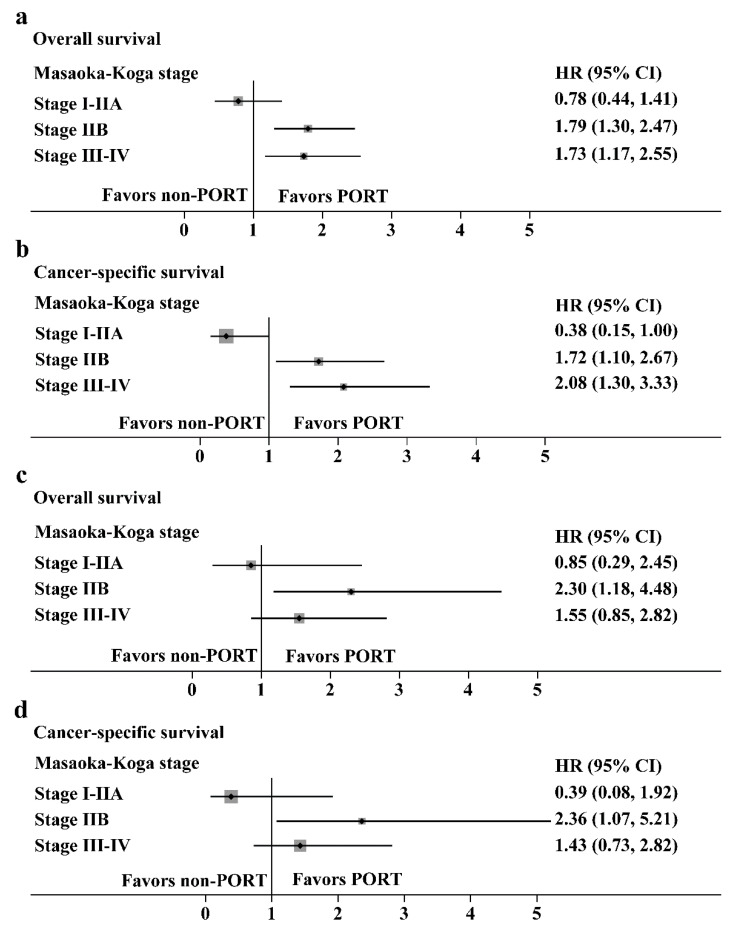
Subgroup analysis with the Cox regression model. (**a**,**b**) Hazard ratios with 95% CI for the overall survival and cancer-specific survival in thymoma stratified by the Masaoka-Koga stage. (**c**,**d**) Hazard ratios with 95% CI for the overall survival and cancer-specific survival in thymic carcinoma stratified by the Masaoka-Koga stage.

**Table 1 cancers-14-04938-t001:** Characteristics of thymoma and thymic carcinoma in the SEER database.

Variables		Number of Patients (%)	
		Thymoma	Thymic Carcinoma
		*n* = 2138	*n* = 420
Age (years)			
	Mean ± SD	59.7 ± 14.1	63.1 ± 12.9
	≤60	1017 (47.6)	161 (38.3)
	>60	1121 (52.4)	259 (61.7)
Gender			
	Male	1156 (54.1)	269 (64.0)
	Female	982 (45.9)	151 (36.0)
Race			
	White	1428 (66.8)	286 (68.1)
	Black	287 (13.4)	58 (13.8)
	Other	397 (18.6)	73 (17.4)
	Unknown	26 (1.2)	3 (0.7)
Year of diagnosis			
	2010-2014	971 (45.4)	179 (42.6)
	2015-2019	1167 (54.6)	241 (57.4)
Other malignancies			
	No	1601 (74.9)	291 (69.3)
	Yes	537 (25.1)	129 (30.7)
Time to treatment (months)			
	≤1	1688 (79.0)	220 (52.4)
	>1	450 (21.0)	200 (47.6)
Masaoka–Koga stage			
	I-IIA	810 (37.9)	99 (23.6)
	IIB	962 (45.0)	219 (52.1)
	III-IV	307 (14.4)	87 (20.7)
	Unknown	59 (2.7)	15 (3.6)
Tumor size (cm)			
	<6.5 (Thymoma)	1020 (47.7)	-
	≥6.5 (Thymoma)	1118 (52.3)	-
	<6.0 (Thymic Carcinoma)	-	198 (47.1)
	≥6.0 (Thymic Carcinoma)	-	222 (52.9)
Lymph Node Dissection			
	No	1165 (54.5)	179 (42.6)
	Yes	933 (43.6)	236 (56.2)
	Unknown	40 (1.9)	5 (1.2)
Extent of surgery			
	Total/radical resection	1247 (58.3)	218 (51.9)
	Local excision/partial removal	798 (37.3)	173 (41.2)
	Debulking/NOS	93 (4.4)	29 (6.9)
WHO classification			
	Type A	206 (9.6)	-
	Type AB	502 (23.5)	-
	Type B1	279 (13.0)	-
	Type B2	410 (19.2)	-
	Type B3	365 (17.1)	-
	NOS	376 (17.6)	-
Grade			
	Well	95 (4.4)	10 (2.4)
	Moderate	62 (2.9)	30 (7.1)
	Poor/Undifferentiated	189 (8.9)	105 (25.0)
	Unknown	1792 (83.8)	275 (65.5)
PORT			
	Yes	963 (45.0)	168 (40.0)
	No	1175 (55.0)	252 (60.0)

Abbreviations: SD, standard deviation; PORT, postoperative radiotherapy; WHO, World Health Organization; NOS, not otherwise specified. Data are presented as *n* (%).

**Table 2 cancers-14-04938-t002:** Characteristics of thymoma patients before and after propensity score matching.

Characteristics		Entire Population					Propensity-Matched Popuplation				
		PORT (−)		PORT (+)		*p*-Value	PORT (−)		PORT (+)		*p*-Value
		(*n* = 1175)	%	(*n* = 963)	%		(*n* = 783)	%	(*n* = 783)	%	
Age (years)						0.166					1.000
	≤60	543	46.2	474	49.2		369	47.1	369	47.1	
	>60	632	53.8	489	50.8		414	52.9	414	52.9	
Gender						0.034					0.648
	Male	611	52.0	545	56.6		424	54.2	415	53.0	
	Female	564	48.0	418	43.4		359	45.8	368	47.0	
Race						0.088					0.458
	White	799	68.8	629	66.2		509	65.8	527	67.9	
	Black	164	14.1	123	12.9		101	13.1	105	13.5	
	Other	199	17.1	198	20.8		163	21.1	144	18.6	
	Unknown	13	-	13	-		10	-	7	-	
Year of diagnosis						0.234					0.648
	2010–2014	520	44.3	451	46.8		368	47.0	359	45.8	
	2015–2019	655	55.7	512	53.2		415	53.0	424	54.2	
Other malignancies						0.028					0.383
	No	858	73.0	743	77.2		578	73.8	593	75.7	
	Yes	317	27.0	220	22.8		205	26.2	190	24.3	
Time to treatment (months)						0.354					0.755
	≤1	919	78.2	769	79.9		619	79.1	624	79.7	
	>1	256	21.8	194	20.1		164	20.9	159	20.3	
Masaoka–Koga stage						<0.001					0.633
	I-IIA	580	50.8	230	24.5		240	31.5	230	30.3	
	IIB	413	36.2	549	58.5		397	52.0	389	51.3	
	III-IV	148	13.0	159	17.0		126	16.5	139	18.4	
	Unknown	34	-	25	-		20	-	25	-	
Tumor size (cm)						0.765					0.814
	<6.5	564	48.0	456	47.4		370	47.3	373	47.6	
	≥6.5	611	52.0	507	52.6		413	52.7	410	52.4	
Lymph Node Dissection						<0.001					0.599
	No	686	59.7	479	50.5		424	54.9	411	53.5	
	Yes	464	40.3	469	49.5		349	45.1	357	46.5	
	Unknown	25	-	15	-		10	-	15	-	
Extent of surgery						<0.001					0.617
	Total/radical resection	616	52.4	631	65.5		461	58.9	467	59.6	
	Local excision/partial removal	510	43.4	288	29.9		287	36.7	272	34.8	
	Debulking/NOS	49	4.2	44	4.6		35	4.4	44	5.6	
WHO classification						<0.001					0.542
	Type A	140	14.4	66	8.4		68	11.1	66	10.8	
	Type AB	315	32.4	187	23.7		165	27.1	164	26.8	
	Type B1	165	16.9	114	14.5		109	17.9	96	15.7	
	Type B2	199	20.4	211	26.8		146	23.9	152	24.8	
	Type B3	155	15.9	210	26.6		122	20.0	134	21.9	
	NOS	201	-	175	-		173	-	171	-	
Grade						0.063					0.881
	Well	51	34.2	44	22.3		37	29.4	33	26.2	
	Moderate	27	18.1	35	17.8		21	16.7	21	16.7	
	Poor/Undifferentiated	71	47.7	118	59.9		68	53.9	72	57.1	
	Unknown	1026	-	766	-		657	-	657	-	

Abbreviations: PORT, postoperative radiotherapy; WHO, World Health Organization; NOS, not otherwise specified.

**Table 3 cancers-14-04938-t003:** Characteristics of thymic carcinoma patients before and after propensity score matching.

Characteristics		Entire Population					Propensity-Matched Popuplation				
		PORT (−)		PORT (+)		*p*-Value	PORT (−)		PORT (+)		*p*-Value
		(*n* = 168)	%	(*n* = 252)	%		(*n* = 156)	%	(*n* = 156)	%	
Age (years)						0.130					0.633
	≤60	57	33.9	104	41.3		55	35.3	51	32.7	
	>60	111	66.1	148	58.7		101	64.7	105	67.3	
Gender						0.340					0.815
	Male	103	61.3	166	65.9		97	62.2	99	63.5	
	Female	65	38.7	86	34.1		59	37.8	57	36.5	
Race						0.851					0.576
	White	115	69.3	171	68.2		106	68.4	108	69.7	
	Black	24	14.5	34	13.5		22	14.2	26	16.8	
	Other	27	16.2	46	18.3		27	17.4	21	13.5	
	Unknown	2	-	1	-		1	-	1	-	
Year of diagnosis						0.468					0.425
	2010–2014	68	40.5	111	44.0		66	42.3	73	46.8	
	2015–2019	100	59.5	141	56.0		90	57.7	83	53.2	
Other malignancies						0.342					0.341
	No	112	66.7	179	71.0		106	67.9	98	62.8	
	Yes	56	33.3	73	29.0		50	32.1	58	37.2	
Time to treatment (months)						0.231					1.000
	≤1	94	56.0	126	50.0		84	53.8	84	53.8	
	>1	74	44.0	126	50.0		72	46.2	72	46.2	
Masaoka–Koga stage						0.005					0.928
	I-IIA	47	29.2	52	21.3		43	28.9	43	28.7	
	IIB	71	44.1	148	60.7		71	47.7	69	46.0	
	III-IV	43	26.7	44	18.0		35	23.5	38	25.3	
	Unknown	7	-	8	-		7	-	6	-	
Tumor size (cm)						0.216					0.762
	<6.0	73	43.5	125	49.6		61	39.1	59	37.8	
	≥6.0	95	56.5	127	50.4		95	60.9	97	62.2	
Lymph Node Dissection		168		252		0.021					0.732
	No	83	50.0	96	38.6		75	48.7	72	46.8	
	Yes	83	50.0	153	61.4		79	51.3	82	53.2	
	Unknown	2	-	3	-		2	-	2	-	
Extent of surgery						0.029					0.893
	Total/radical resection	75	44.7	143	56.7		67	42.9	70	44.9	
	Local excision/partial removal	77	45.8	96	38.1		74	47.5	73	46.8	
	Debulking/NOS	16	9.5	13	5.2		15	9.6	13	8.3	
Grade						0.503					0.854
	Well	3	5.5	7	7.8		3	5.6	4	6.7	
	Moderate	14	25.5	16	17.8		14	25.9	13	21.7	
	Poor/Undifferentiated	38	69.0	67	74.4		37	68.5	43	71.6	
	Unknown	113	-	162	-		102	-	96	-	

Abbreviations: PORT, postoperative radiotherapy; NOS, not otherwise specified.

**Table 4 cancers-14-04938-t004:** Univariate and Multivariate Cox regression analysis of clinical characteristics for overall survival and cancer-specific survival rate in thymoma patients in the matched population.

Characteristics		OS								CSS							
		Univariate Analysis	95%CI Lower	95%CI Upper	*p*-Value	Multivariate Analysis	95%CI Lower	95%CI Upper	*p*-Value	Univariate Analysis	95%CI Lower	95%CI Upper	*p*-Value	Multivariate Analysis	95%CI Lower	95%CI Upper	*p*-Value
Age (years)																	
	≤60	1	-	-		1	-	-		1	-	-		1	-	-	
	>60	1.956	1.551	2.467	<0.001	2.007	1.673	2.711	<0.001	1.345	1.001	1.808	0.049	1.426	1.173	1.941	0.034
Gender																	
	Male	1	-	-		1	-	-		1	-	-		1	-	-	
	Female	1.343	1.074	1.679	0.010	1.312	1.019	1.630	0.011	1.351	1.003	1.819	0.048	1.274	0.933	1.643	0.109
Race																	
	White	1	-	-						1	-	-					
	Black	1.186	0.856	1.643	0.305					0.991	0.622	1.579	0.971				
	Other	1.158	0.879	1.526	0.297					1.309	0.922	1.858	0.132				
Year of diagnosis																	
	2010–2014	1	-	-						1	-	-					
	2015–2019	0.875	0.666	1.150	0.337					0.929	0.660	1.307	0.671				
Other malignancies																	
	No	1	-	-		1	-	-		1	-	-					
	Yes	1.245	0.981	1.581	0.072	1.069	0.803	1.417	0.407	0.779	0.546	1.111	0.168				
Time to treatment (months)																	
	≤1	1	-	-						1	-	-					
	>1	0.774	0.571	1.050	0.110					0.773	0.517	1.157	0.211				
Masaoka–Koga stage																	
	I-IIA	1	-	-		1	-	-		1	-	-		1	-	-	
	IIB	2.342	1.682	3.260	<0.001	2.246	1.631	3.116	<0.001	2.733	1.675	4.448	<0.001	2.442	1.545	3.986	<0.001
	III-IV	4.829	3.403	6.854	<0.001	4.631	3.192	6.524	<0.001	7.461	4.547	12.238	<0.001	6.461	3.683	10.112	<0.001
Tumor size (cm)																	
	<6.5	1	-	-		1	-	-		1	-	-		1	-	-	
	≥6.5	1.725	1.339	2.141	<0.001	1.586	1.128	2.049	<0.001	2.429	1.651	2.728	<0.001	1.719	1.328	2.351	<0.001
Lymph Node Dissection																	
	No	1	-	-						1	-	-		1	-	-	
	Yes	1.153	0.922	1.442	0.212					1.373	1.020	1.847	0.036	1.015	0.706	1.429	0.714
Extent of surgery																	
	Total/radical resection	1	-	-		1	-	-		1	-	-					
	Local excision/partial removal	1.283	0.974	1.69	0.076	1.538	0.506	1.892	0.295	0.803	0.486	1.328	0.393				
	Debulking/NOS	1.373	0.773	2.438	0.279	1.882	0.843	4.200	0.123	1.826	0.823	4.050	0.139				
WHO classification																	
	Type A	1	-	-						1	-	-					
	Type AB	0.722	0.401	1.301	0.279					0.42	0.164	1.152	0.214				
	Type B1	0.819	0.435	1.543	0.536					0.639	0.224	1.824	0.403				
	Type B2	1.500	0.864	2.605	0.150					1.483	0.627	3.508	0.370				
	Type B3	1.620	0.914	2.879	0.109					1.75	0.756	4.052	0.191				
Grade																	
	Well	1	-	-		1	-	-		1	-	-		1	-	-	
	Moderate	2.174	1.095	4.315	0.026	2.282	1.415	3.724	<0.001	3.211	1.805	5.711	<0.001	2.926	1.682	5.311	<0.001
	Poor/Undifferentiated	2.543	1.232	3.853	<0.001	2.586	1.867	3.273	<0.001	4.499	3.044	5.954	<0.001	3.812	2.680	5.504	<0.001
PORT																	
	Yes	1	-	-		1	-	-		1	-	-		1	-	-	
	No	1.482	1.186	1.852	<0.001	1.627	1.337	2.927	<0.001	1.395	1.038	1.873	0.026	1.598	1.224	2.185	0.001

Abbreviations: OS, overall survival; CSS, cancer-specific survival; CI, confidence interval; PORT, postoperative radiotherapy; WHO, World Health Organization; NOS, not otherwise specified.

**Table 5 cancers-14-04938-t005:** Univariate and Multivariate Cox regression analysis of clinical characteristics for overall survival and cancer-specific survival rate in thymic carcinoma patients in the matched population.

Characteristics		OS								CSS							
		Univariate Analysis	95%CI Lower	95%CI Upper	*p*-Value	Multivariate Analysis	95%CI Lower	95%CI Upper	*p*-Value	Univariate Analysis	95%CI Lower	95%CI Upper	*p*-Value	Multivariate Analysis	95%CI Lower	95%CI Upper	*p*-Value
Age (years)																	
	≤60	1	-	-						1	-	-					
	>60	0.978	0.648	1.475	0.915					0.949	0.587	1.532	0.829				
Gender																	
	Male	1	-	-						1	-	-					
	Female	1.026	0.685	1.537	0.900					1.000	0.623	1.603	0.999				
Race																	
	White	1	-	-						1	-	-					
	Black	0.841	0.456	1.553	0.581					0.868	0.426	1.766	0.695				
	Other	1.345	0.801	2.260	0.262					1.445	0.798	2.615	0.224				
Year of diagnosis																	
	2010–2014	1	-	-						1	-	-					
	2015–2019	0.811	0.537	1.224	0.318					0.844	0.524	1.360	0.486				
Other malignancies																	
	No	1	-	-						1	-	-		1	-	-	
	Yes	0.767	0.500	1.177	0.225					0.636	0.377	1.073	0.090	0.827	0.403	1.465	0.328
Time to treatment (months)																	
	≤1	1	-	-						1	-	-					
	>1	1.432	0.966	2.121	0.074	1.164	0.831	1.867	0.274	1.393	0.879	2.207	0.158				
Masaoka–Koga stage																	
	I-IIA	1	-	-		1	-	-		1	-	-		1	-	-	
	IIB	2.068	1.123	3.809	0.020	2.085	1.171	3.836	0.012	2.581	1.176	5.664	0.018	2.694	1.212	5.773	0.011
	III-IV	4.897	2.675	8.964	<0.001	4.829	2.603	8.795	<0.001	6.635	3.067	14.352	<0.001	6.679	3.165	14.447	<0.001
Tumor size (cm)																	
	<6.0	1	-	-		1	-	-		1	-	-		1	-	-	
	≥6.0	1.682	1.158	2.443	0.006	1.386	0.943	2.037	0.097	2.066	1.312	3.255	0.002	1.724	1.086	2.738	0.021
Lymph Node Dissection																	
	No	1	-	-						1	-	-					
	Yes	2.284	0.551	9.470	0.255					1.372	0.856	2.197	0.189				
Extent of surgery																	
	Total/radical resection	1	-	-		1	-	-		1	-	-		1	-	-	
	Local excision/partial removal	1.554	1.060	2.278	0.024	1.470	1.001	2.160	0.050	1.924	1.217	3.043	0.005	1.851	1.168	2.934	0.009
	Debulking/NOS	3.100	1.782	5.394	<0.001	2.927	1.679	5.103	<0.001	3.391	1.738	6.616	<0.001	3.252	1.663	6.36	0.001
Grade																	
	Well	1	-	-						1	-	-					
	Moderate	2.031	0.244	16.872	0.512					1.689	0.197	14.460	0.632				
	Poor/Undifferentiated	4.555	0.624	33.249	0.135					3.910	0.533	28.666	0.180				
PORT																	
	Yes	1	-	-		1	-	-		1	-	-					
	No	1.435	0.967	2.131	0.073	1.593	1.085	2.401	0.018	1.279	0.807	2.027	0.294				

Abbreviations: OS, overall survival; CSS, cancer-specific survival; CI, confidence interval; PORT, postoperative radiotherapy; NOS, not otherwise specified.

## Data Availability

The data in this study can be obtained from the public SEER database (www.seer.cancer.gov (accessed on 16 August 2022)). Permission to access the research data file in the SEER registry was received from the NCI, USA (reference No. 16521-Nov 2021).

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
