# Peer review of "The Prognostic Value of Postoperative Radiotherapy for Thymoma and Thymic Carcinoma: A Propensity-Matched Study Based on SEER Database"

_cancers, 2022, doi:10.3390/cancers14194938_

Round 1
Reviewer 1 Report
The autors report a retrospective study about the impact of PORT in thymomas and thymic carcinomas using a propensity score matched analysis of patients taken from SEER database. The paper is well written and the used methodology aiming to reduce the retrospective limititation of analysis is The methodology used appears to be very suitable. The results demonstrate an advantage in terms of OS and CSS for PORT in patients with stage IIb - III - IV. I believe that one of the strengths of the work is the large number of patients. However one of the matching variables does not appear to be the degree of surgical resection. The literature shows a fundamental impact on the degree of resection in determining the advantage of the PORT in terms of OS and CSS, especially in stages IIB and III. I think it is important to report this limit of the study or, better yet, it would be appropriate to carry out a matching considering the degree of resection.
Author Response
Comment 1: The literature shows a fundamental impact on the degree of resection in determining the advantage of the PORT in terms of OS and CSS, especially in stages IIB and III. I think it is important to report this limit of the study or, better yet, it would be appropriate to carry out a matching considering the degree of resection.
Reply 1: We accepted your suggestion and extracted data on the extent of surgical resection for analysis.

Reviewer 2 Report
Thanks for the opportunity of reviewing this manuscript which retrospectively investigated how postoperative radiotherapy impacted the outcomes of thymoma and thymic carcinoma.
With regards to time to treatment, why was one month selected as the cutoff value? Similarly, why did the authors use 7.4 cm as the cutoff value for tumor size?
In Line 137-140, For patients with thymoma, the PORT group had a better OS than non-PORT group (survival rates during the follow-up period were respectively 82.5 and 79.5%, p = 0.021; Figure 1a), but PORT was not associated with CSS before matching (89.4 and 90.5%, p = 0.819; Figure 1b). For patients with thymic carcinoma, both OS (76.2 and 64.3%, p = 0.002; Figure 1c) and CSS (82.1 and 75.0%, p = 0.030; Figure 1d) of PORT group are better than non-PORT group. What were the timepoints for the reported survival rates?
Some discordant points were noted. In the section of “Statistical Analysis”, it was mentioned that univariate Cox regression analysis was used to determine variables associated with OS and CSS of the matched population, and variables with P value less than 0.05 were selected for multivariate Cox regression model. However, in table 4 for OS of thymoma, other malignancy with p value of 0.072 in univariate analysis was selected for multivariate Cox regression model. Similarly, in table 5 for OS of thymic carcinoma, time to treatment with p value of 0.074 and PORT with p value of 0.073 in univariate analysis were selected for multivariate Cox regression model. Additionally, in table 5 for CSS of thymic carcinoma, other malignancy with p value of 0.090 in univariate analysis was selected for multivariate Cox regression model. These variables were included in multivariate Cox regression model although they did not have P value less than 0.05.
In Line 231-232, when there was no significant difference, is it proper to state the survival rate of patients with stage III-IV thymic carcinoma in the PORT group was better (OS: 44.7% vs. 37.1%; CSS: 232 55.3% vs. 51.4%)?
In Line 239-240, it was mentioned that many studies reported a slight predominance of 239 women with type A, AB, and B1, which might be one of the reasons. The relevant reference should be added.
In Line 248-249, it was mentioned that we did not analyze the prognostic value of PORT between different surgical methods and different histological classifications. Is the information about surgical methods and histological classifications included in the SEER database? If the above information is included in the SEER database, the relevant analyses need to be done.
Figure 1 legend needs to be improved.
Author Response
Comment 1: With regards to time to treatment, why was one month selected as the cutoff value? Similarly, why did the authors use 7.4 cm as the cutoff value for tumor size?
Reply 1: Thank you for your rigorous consideration. In the SEER database, the time unit from diagnosis to treatment is month, and most patients received treatment within one month after diagnosis, so the cutoff value was set at one month. Originally, we calculated the cutoff value based on the mean tumor size. Based on your suggestion, we calculated the median tumor size for thymoma and thymoma separately and set it as the new cutoff value (line 120-121).
Comment 2: In Line 137-140, For patients with thymoma, the PORT group had a better OS than non-PORT group (survival rates during the follow-up period were respectively 82.5 and 79.5%, p = 0.021; Figure 1a), but PORT was not associated with CSS before matching (89.4 and 90.5%, p = 0.819; Figure 1b). For patients with thymic carcinoma, both OS (76.2 and 64.3%, p = 0.002; Figure 1c) and CSS (82.1 and 75.0%, p = 0.030; Figure 1d) of PORT group are better than non-PORT group. What were the timepoints for the reported survival rates?
Reply 2: Thanks for your comment. Survival rates in this article are calculated based on the entire follow-up time (94.1 months (95% CI, 92.1 - 96.4 months) for patients with thymoma and 80.8 months (95% CI, 75.4 - 86.1 months) for patients with thymic carcinoma).
Comment 3: Some discordant points were noted. In the section of “Statistical Analysis”, it was mentioned that univariate Cox regression analysis was used to determine variables associated with OS and CSS of the matched population, and variables with P value less than 0.05 were selected for multivariate Cox regression model. However, in table 4 for OS of thymoma, other malignancy with p value of 0.072 in univariate analysis was selected for multivariate Cox regression model. Similarly, in table 5 for OS of thymic carcinoma, time to treatment with p value of 0.074 and PORT with p value of 0.073 in univariate analysis were selected for multivariate Cox regression model. Additionally, in table 5 for CSS of thymic carcinoma, other malignancy with p value of 0.090 in univariate analysis was selected for multivariate Cox regression model. These variables were included in multivariate Cox regression model although they did not have P value less than 0.05.
Reply 3: We have corrected the error in the text (p < 0.1), for which we apologize (line 111).
Comment 4: In Line 231-232, when there was no significant difference, is it proper to state the survival rate of patients with stage III-IV thymic carcinoma in the PORT group was better (OS: 44.7% vs. 37.1%; CSS: 232 55.3% vs. 51.4%)?
Reply 4: We gratefully appreciate for your suggestion. We have revised this sentence in the manuscript (line 239-241).
Comment 5:In Line 239-240, it was mentioned that many studies reported a slight predominance of 239 women with type A, AB, and B1, which might be one of the reasons. The relevant reference should be added.
Reply 5: We have added references to the text (line 248).
Comment 6: In Line 248-249, it was mentioned that we did not analyze the prognostic value of PORT between different surgical methods and different histological classifications. Is the information about surgical methods and histological classifications included in the SEER database? If the above information is included in the SEER database, the relevant analyses need to be done.
Reply 6: We gratefully appreciate for your suggestion. As suggested by you and another reviewer, we have included the extent of surgical resection and WHO classification in the analysis.
Comment 7: Figure 1 legend needs to be improved.
Reply 7: We have changed the Figure 1 legend (line 151-152).

Round 2
Reviewer 2 Report
In table 4 & 5 of this revised manuscript, extent of surgery was additionally taken into consideration in multivariate analyses. I surprisingly found that the HR and CI (in multivariate analyses) of age, gender, other malignancy, grade, and PORT did not differ from those of table 4 & 5 of the first draft. These conditions are not reasonable.
In Line 143-149, In the overall cohort, the average follow-up periods were 94.1 months (95% CI, 92.1 - 96.4 months, Kaplan–Meier 143 estimate) for thymoma and 80.8 months (95% CI, 75.4 - 86.1 months) for thymic carcinoma. The Kaplan-Meier survival 144 curves of the overall cohort according to the receipt of PORT are shown in Figure 1. For patients with thymoma, the 145 PORT group had a better OS than non-PORT group (survival rates during the follow-up period were respectively 82.5 146 and 79.5%, p = 0.021; Figure 1a), but PORT was not associated with CSS before matching (89.4 and 90.5%, p = 0.819; 147 Figure 1b). For patients with thymic carcinoma, both OS (76.2 and 64.3%, p = 0.002; Figure 1c) and CSS (82.1 and 75.0%, 148 p = 0.030; Figure 1d) of PORT group are better than non-PORT group.
Thanks for the authors’ reply, and it was mentioned that survival rates in this article are calculated based on the entire follow-up time (94.1 months (95% CI, 92.1 - 96.4 months) for patients with thymoma and 80.8 months (95% CI, 75.4 - 86.1 months) for patients with thymic carcinoma). But when I look at Fig. 1a, the survival rate at 94.1 months for thymoma cases with PORT was less than 80%. The value was different from 82.5%. Similar conditions were also found for other figures.
In Line 247, it was mentioned that although we did not analyze histological classifications, but authors already included histological classifications in the analysis of current version.
For survival curves, please include the number-at-risk at each time point and identify when patients were censored on each curve.
Author Response
Comment 1: In table 4 & 5 of this revised manuscript, extent of surgery was additionally taken into consideration in multivariate analyses. I surprisingly found that the HR and CI (in multivariate analyses) of age, gender, other malignancy, grade, and PORT did not differ from those of table 4 & 5 of the first draft. These conditions are not reasonable.
Reply 1: We apologize for the errors in the table data. Previously the data processor uploaded the wrong form, we have uploaded the revised form.
Comment 2: Thanks for the authors’ reply, and it was mentioned that survival rates in this article are calculated based on the entire follow-up time (94.1 months (95% CI, 92.1 - 96.4 months) for patients with thymoma and 80.8 months (95% CI, 75.4 - 86.1 months) for patients with thymic carcinoma). But when I look at Fig. 1a, the survival rate at 94.1 months for thymoma cases with PORT was less than 80%. The value was different from 82.5%. Similar conditions were also found for other figures.
Reply 2: Thank you for your rigorous consideration. The data processor previously provided survival rates for the different groups during their respective follow-up periods. For more normalization, we reanalyzed the data and calculated the 5-year survival rate for each group.
Comment 3: In Line 247, it was mentioned that although we did not analyze histological classifications, but authors already included histological classifications in the analysis of current version.
Reply 3: We apologize for this oversight and have revised this in the revised manuscript.
Comment 4: For survival curves, please include the number-at-risk at each time point and identify when patients were censored on each curve.
Reply 4: We appreciate it very much for this good suggestion, and we have done it according to your ideas.

Round 3
Reviewer 2 Report
NA
Author Response
Dear Reviewer
Thanks very much for taking your time to review this manuscript. Thank you again for your comments and suggestions!